Microbiology **Spectrum**

# Serum hepatitis B core antibody as the prognostic factor for diffuse large B-cell lymphoma

Yi Rong,[1,2] Ming Wang,[1] Yaqiong Ma,[1] Yuanchen Liang,[1] Lvyin Ye,[1] Lin Guo,[1,2] Renquan Lu,[1,2] Yanchun Wang[1,2]

**ABSTRACT**  Diffuse large B-cell lymphoma (DLBCL) is the most common subtype of non-Hodgkin lymphoma (NHL), strongly associated with viral infections. Although the link between hepatitis B virus (HBV) infection and DLBCL is well-documented, effective clinical markers reflecting HBV-associated DLBCL remain scarce. This study aims to identify prognostic indicators for HBV-associated DLBCL through retrospective analysis of the relationship among tissue marker molecules, HBV serum markers, and clinical prognosis in DLBCL patients. Here, we found the results that DLBCL patients who tested positive for hepatitis B core antibody (HBcAb) had significantly reduced overall survival (OS) rates compared with those who tested negative. Additionally, a strong correlation was observed between an elevated HBcAb-positive rate and reduced expression of the CD23 molecule in DLBCL tissue samples. Stratifying DLBCL patients based on combined HBcAb-CD23 status revealed significant disparities in OS rates. Therefore, integrating CD23 with HBcAb could be applied to prognostic assessments for individuals with HBV-associated DLBCL. This study identifies novel indicators and diagnostic strategies for HBV-associated DLBCL.

**IMPORTANCE**  This study identifies hepatitis B core antibody (HBcAb) as a significant prognostic indicator for hepatitis B virus (HBV)-associated diffuse large B-cell lymphoma (DLBCL). The findings reveal that patients with DLBCL with positive HBcAb have significantly reduced overall survival rates. Additionally, a strong negative correlation is observed between serum HBcAb and the expression of the CD23 molecule in DLBCL tissues. These results highlight the potential of integrating HBcAb and CD23 as prognostic markers in clinical assessments of HBV-associated DLBCL, offering new insights for risk stratification and treatment planning in this patient population.

**KEYWORDS**   hepatitis B virus, diffuse large B-cell lymphoma, hepatitis B core antibody, CD23, prognosis

Hepatitis B virus (HBV) infection is a global public health issue, affecting nearly one billion people worldwide (1). It is a significant cause of morbidity and mortality, including acute and chronic hepatitis and hepatocellular carcinoma (HCC) (2). Although HBV primarily affects liver disorders, recent studies suggest that it may also be associated with other types of cancers outside the liver. HBV infection is linked to DLBCL, with higher positivity rates in Asia (e.g., 30%–40% in Southern China) and sub-Saharan Africa (1). In Western countries, overall prevalence is lower, but immigrants from endemic regions may have higher rates (3). HBV can cause a chronic inflammatory response (4–6), and several studies have demonstrated HBV's ability to infect human lymphocytes (7, 8), although our understanding of this process remains limited. Oncogenic HBV variants and integration within liver cells and lymphocytes are found in patients with chronic HBV infection. Samples from chronic HBV carriers revealed that HBV replicates not only in the liver but also in lymphocytes, with replication enhanced *in vitro*-stimulated peripheral

**Peer Reviewers** Mushtak T. S. Al-Ouqaili, University of Anbar, Ramadi, Al-Anbar, Iraq; Mai Abdel Haleem A. Abusalah, Universiti Sains Malaysia, Kota Bharu, Kelantan, Malaysia

Address correspondence to Lin Guo, guolin500@hotmail.com, Renquan Lu, renquanlu@fudan.edu.cn, or Yanchun Wang, wang_yanchun@fudan.edu.cn.

The authors declare no conflict of interest.

See the funding table on p. 9.

blood mononuclear cells (PBMCs), potentially contributing to the pathogenesis of B-cell lymphomas (9). Chronic HBV infection increases the risk of developing DLBCL through immune dysfunction and viral DNA integration. Persistent immune activation impairs the clearance of abnormal B cells, promoting lymphoma development (10). HBV-DNA integration into the host genome can cause gene mutations, activating antiapoptotic mechanisms and enhancing DLBCL malignancy (11). Clinical risk factors include HBsAg positivity and high HBV-DNA levels (12). The finding of HBV-encoded oncogene X (*HBx*) integration in diffuse large B-cell lymphoma (DLBCL), the most common non-Hodgkin lymphoma (NHL) subtype, also sheds light on HBV's role in tumorigenesis and the development of non-hepatic malignancies (13).

A meta-analysis of 58 studies supports a positive link between HBV infection and NHL development, with a summary odds ratio (sOR) of 2.50, indicating a significantly increased NHL risk in individuals with HBV infection (14). DLBCL is highly heterogeneous and closely related to viral infections and immune status (15). HBV-associated DLBCL patients tend to have more advanced clinical stages with distinct clinical features of poor chemotherapy response compared with their non-HBV counterparts (9). The relationship between HBV and DLBCL is further complicated by the fact that patients with DLBCL who have a history of HBV infection are at an increased risk for HBV reactivation following chemotherapy and immunotherapy. This reactivation can lead to adverse clinical outcomes and poor prognosis, necessitating close monitoring and preemptive antiviral treatment (16). Moreover, the clinical features of HBV-related DLBCL are distinct, often presenting with more advanced disease and higher international prognostic index (IPI) scores, along with increased levels of lactate dehydrogenase (LDH) etc. (9, 17). Elevated LDH levels indicate higher tumor burden and aggressive disease (18). The prognosis for HBV-related DLBCL is generally poor (19), highlighting the need for effective treatment strategies and a deeper understanding of the mechanisms by which HBV contributes to lymphomagenesis.

At present, the prognostic indicators for DLBCL predominantly encompass the IPI and progression-free survival (PFS) among other metrics (20, 21). Nevertheless, these indicators exhibit certain constraints, such as an incapacity for dynamic assessment and a propensity to either underestimate or overestimate the prognosis in specific patient cohorts. One of the main challenges is the heterogeneity of DLBCL itself (22), which is further complicated by the interaction with HBV. The molecular basis of this heterogeneity is beginning to be understood through gene-expression profiling, which divides DLBCL into different subtypes such as the activated B-cell-like (ABC) and germinal center B-cell-like (GCB) (23). However, these classifications do not always provide clear prognostic indicators for HBV-associated cases. Although the pathogenesis of lymphoma is related to HBV antigens and HBV-DNA (24), scarcely any literature has reported the relationship between HBV antibodies and DLBCL. Therefore, we conducted a comprehensive study on the relationship between the HBV status of DLBCL patients and clinical prognosis to promote a comprehensive understanding of the association between HBV infection and the development of DLBCL.

## MATERIALS AND METHODS

### Patient recruitment

A comprehensive retrospective study was conducted at Fudan University Shanghai Cancer Center, encompassing a total of 4,491 patients diagnosed with DLBCL from January 2015 to January 2024. The cohort consisted of 4,491 patients diagnosed with DLBCL with a median age of 51 years (interquartile range of 35–64 years), with a male predominance (male:female ratio of 1.5:1). All participants received standard histological diagnosis and treatment protocols during their course of care at the institution. Patients included in this study were diagnosed with DLBCL and received standard first-line chemotherapy regimens, such as CHOP/R-CHOP (rituximab, cyclophosphamide, doxorubicin, vincristine, and prednisone). Patients with a history of prior malignancies

or those who had received non-standard therapeutic interventions were excluded. Additionally, only patients with confirmed HBV infection status (either positive or negative for HBV indicators) were included to ensure an accurate analysis of the impact of HBV on DLBCL prognosis. The study meticulously excluded patients with incomplete follow-up data to ensure the accuracy and reliability of the study outcomes. Patients who did not return for clinic visits and did not reserve phone numbers for telephone follow-up were initially excluded. During follow-up, patients who were lost to follow up because they changed their phone number or could not be contacted further for other reasons were censored. The inclusion criteria for this study were stringent, with a focused emphasis on patients with a confirmed diagnosis of DLBCL who had received standard therapeutic regimens. Efforts were made to procure follow-up and monitoring data post-treatment to evaluate long-term outcomes.

## Clinical data collection

Basic information, clinical characteristics, survival status, DLBCL diagnosis and staging, immunohistochemistry (IHC) results, and HBV serological indicators (hepatitis B surface antigen [HBsAg], surface antibody [HBsAb], e antigen [HBeAg], e antibody [HBeAb], core antibody [HBcAb], and HBV-DNA) of the patients were collected. For each patient, 5 mL of blood was collected using a serum separator tube (plasma separator tube for HBV-DNA), followed by centrifugation. Blood samples were collected at the time of initial diagnosis, prior to the initiation of therapeutic intervention in our hospital. This timing maximally ensured that the HBV serological indicators reflected the status not influenced by chemotherapy. HBV series assay kit (Abbott ARCHITECT Alinity) was utilized, employing the chemiluminescent microparticle immunoassay method according to the manufacturer's instructions. HBV-DNA was analyzed by quantitative polymerase chain reaction (PCR) using a commercial kit (Shanghai Haoyuan, China) as per the manufacturer's instructions. Tissue samples were obtained through biopsy and preserved using neutral buffered formalin. The immunohistochemical analysis included the detection of Bcl-2, MUM1, Bcl-6, CyclinD1, CD21, EBER (in situ hybridization), P53, C-myc, CD5, and CD23 proteins. This study retrospectively collected the clinical test results of the patients and was granted by the ethics committee of Fudan University Shanghai Cancer Center for ethical approval (approval number 050432-4-2307E).

## Patient follow-up

Effective follow-ups were conducted on the 1,062 DLBCL patients, with follow-up information obtained through medical records or telephone until January 2024. Overall survival (OS) was defined as the time from disease diagnosis to death or the last follow-up visit.

## Statistical analysis

Patients were grouped based on the results of HBV and IHC. Categorical parameters of patient groups were compared using Pearson's $\chi^2$ test or Fisher's exact test. The Mann-Whitney method was utilized to analyze the non-normally distributed data (such as HBcAb concentration, etc.). Survival curves were calculated using the Kaplan-Meier method, and the Log-rank test was used to identify the differences. Univariate and multivariate analyses to determine prognostic factors affecting survival were performed using the Cox proportional hazards model. A $P$-value less than 0.05 was considered statistically significant. All calculations were performed using SPSS version 26.0, and graphs were illustrated using GraphPad Prism 6.

## RESULTS

### HBV indicators for the prognosis of DLBCL

The results of serum HBV indicators and immunohistochemical staining of DLBCL tissues were collected from enrolled patients with confirmed DLBCL. OS analysis was performed

on 1,062 patients followed up. The results showed that certain serum HBV indicators were associated with OS in patients with DLBCL. Patients with positive HBcAb and HBV-DNA at the antibody level and nucleic acid level had significantly lower 5-year OS than those with negative ones ($P = 0.0202$, $P = 0.0393$, respectively), but there was no significant difference in 5-year OS between HBsAb, HBeAb, HBsAg, and HBeAg groups (Fig. 1). Similar results were obtained in the analysis of OS, with a significantly lower OS in patients with positive HBcAb ($P = 0.0181$), but no significant correlation between OS and the other five HBV indicators (Fig. S1). Therefore, HBcAb was one of the most valuable prognostic indicators of DLBCL patients.

## Tissue protein indicators for the prognosis of DLBCL

The IHC diagnostic parameters of routine DLBCL were analyzed to screen the indicators related to the survival prognosis of DLBCL. The positivity rates for CD5, CD23, Bcl-2, and Bcl-6 were 8.7%, 31.7%, 77.1%, and 89.9%, respectively. It was found that the 5-year OS of DLBCL patients with apoptosis indicator molecule Bcl-2 positive and Bcl-6 negative was significantly reduced ($P = 0.0263$, $P = 0.0393$, respectively, Fig. 2). Similarly, in the OS analysis, Bcl-2 positive was still a poor prognostic factor, whereas different statuses of CD5 also showed significant differences in OS ($P = 0.0145$, $P = 0.0362$, respectively, Fig. S2). However, in the current study, CD23 did not show significant differences in survival of these non-HBV classified DLBCL patients ($P = 0.0951$ Fig. 2, $P = 0.1120$ Fig. S2).

## Correlation between HBV status and tissue protein expression in DLBCL

In order to further screen the molecular indicators of HBV-related DLBCL, we performed a correlation analysis on 4,491 patients between HBV serological indicators and DLBCL histological proteins. The results showed that the serum HBcAb level was most correlated with the expression of the CD23 molecule in tumor tissue (Table 1, the correlation between HBcAb/HBV-DNA and tissue protein expression), and the CD23 positive rate was significantly decreased in HBcAb-positive patients ($P = 0.002$), whereas other conventional histochemical indicators were not significantly correlated with HBcAb, suggesting that CD23 was related to HBV-associated DLBCL with a certain specificity. Among other indicators of HBV status, there was no significant correlation between HBV-DNA and DLBCL protein (Table 1). Except for the correlation between HBcAb and CD23, there was

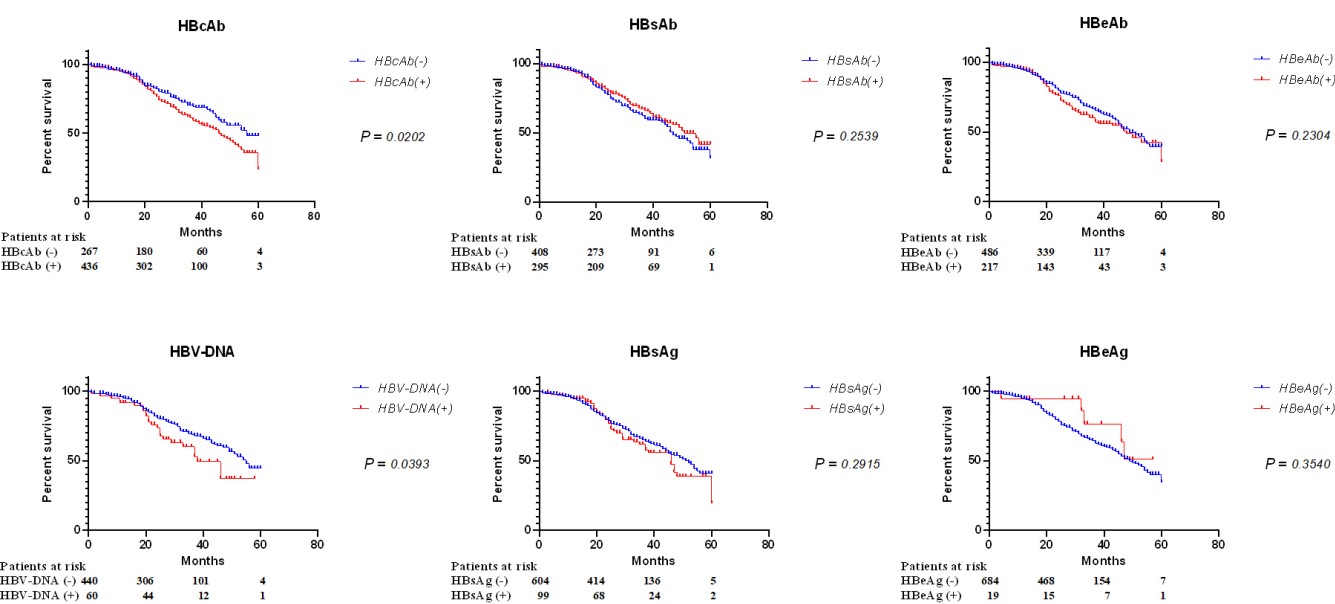

**FIG 1** Five-year OS of DLBCL patients with different serum HBV status. A total of 1,062 DLBCL patients followed up were grouped according to their serum HBV status: HBcAb, HBsAb, HBeAb, HBV-DNA, HBsAg, and HBeAg. The difference in 5-year OS rates among these groups was compared through survival analysis.

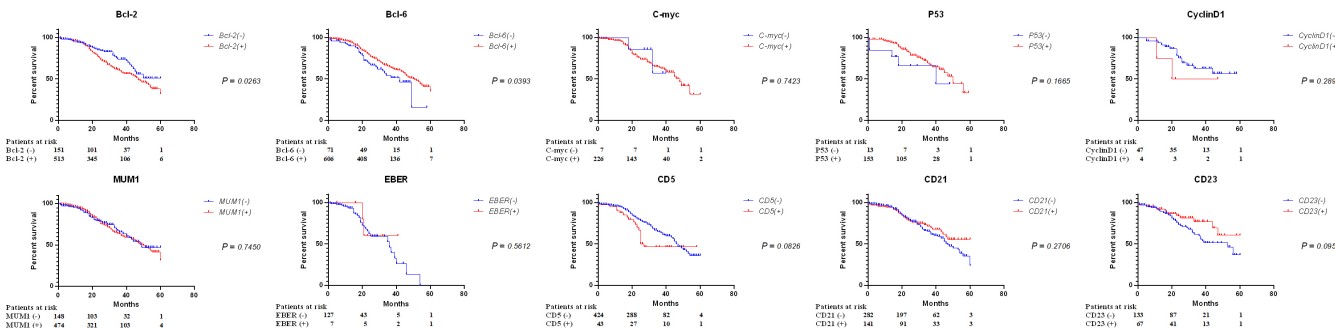

**FIG 2** Five-year OS of DLBCL patients with different protein status in DLBCL tissues. The IHC results of follow-up DLBCL patients were analyzed, and the patients were grouped according to the qualitative status of DLBCL tissue proteins, including Bcl-2, Bcl-6, C-myc, P53, CyclinD1, MUM1, EBER, CD5, CD21, CD23, etc., and the differences in 5-year OS rates between groups were compared through survival analysis.

no significant correlation between other HBV indicators and DLBCL protein expression (Table S1, the correlation between all HBV indicators and tissue protein expression). The Bcl-2 molecule, which is related to the prognosis of DLBCL, did not show a significant correlation with HBV in qualitative analysis (Table 1).

## Combined prognosis analysis of CD23 and HBcAb

Therefore, we further combined the correlation markers of CD23 and HBcAb (r = −0.101, $P$ = 0.003) evaluating the effect on the survival prognosis of DLBCL patients. According to the grouping of DLBCL patients by HBcAb combined with CD23, the 5-year OS rate between groups was significantly different ($P$ = 0.0432) (Fig. 3). Distinct from the results

**TABLE 1** Correlation between HBV status and tissue protein expression in DLBCL[a]

| IHC | | HBV | | | | | |
|---|---|---|---|---|---|---|---|
| | | **HBcAb** | | | **HBV-DNA** | | |
| | | (−) | (+) | $P$ | (−) | (+) | $P$ |
| Bcl-2 | (−) | 272 | 376 | 0.105 | 315 | 38 | 0.169 |
| | (+) | 801 | 1,284 | | 948 | 149 | |
| Bcl-6 | (−) | 133 | 197 | 0.589 | 147 | 23 | 0.723 |
| | (+) | 943 | 1,490 | | 1141 | 164 | |
| C-myc | (−) | 21 | 22 | 0.203 | 26 | 3 | 1.000[b] |
| | (+) | 387 | 602 | | 430 | 56 | |
| P53 | (−) | 17 | 36 | 0.249 | 26 | 2 | 0.558[b] |
| | (+) | 254 | 379 | | 284 | 45 | |
| CyclinD1 | (−) | 95 | 185 | 0.416[b] | 104 | 16 | 1.000[b] |
| | (+) | 3 | 3 | | 6 | 0 | |
| MUM1 | (−) | 255 | 357 | 0.122 | 300 | 54 | 0.088 |
| | (+) | 735 | 1,191 | | 858 | 114 | |
| EBER | (−) | 410 | 605 | 0.381 | 335 | 37 | 0.723[b] |
| | (+) | 32 | 38 | | 21 | 3 | |
| CD5 | (−) | 724 | 1,143 | 0.346 | 859 | 132 | 0.640 |
| | (+) | 70 | 128 | | 79 | 14 | |
| CD21 | (−) | 463 | 753 | 0.900 | 571 | 76 | 0.187 |
| | (+) | 238 | 392 | | 309 | 53 | |
| CD23 | (−) | 241 | 383 | **0.003** | 294 | 43 | 0.998 |
| | (+) | 118 | 119 | | 123 | 18 | |

[a]The association between HBV indicators (HBcAb and HBV-DNA) and various DLBCL tissue molecules in DLBCL patients was compared using the Chi-square test. The significance level of $P$<0.05 was indicated in bold.
[b]When the expected frequency was less than 5, the exact probability method was applied. Due to incomplete histochemistry results for some diagnosed DLBCL patients, the total number of individual histochemistry results was actually ≤4491 cases.

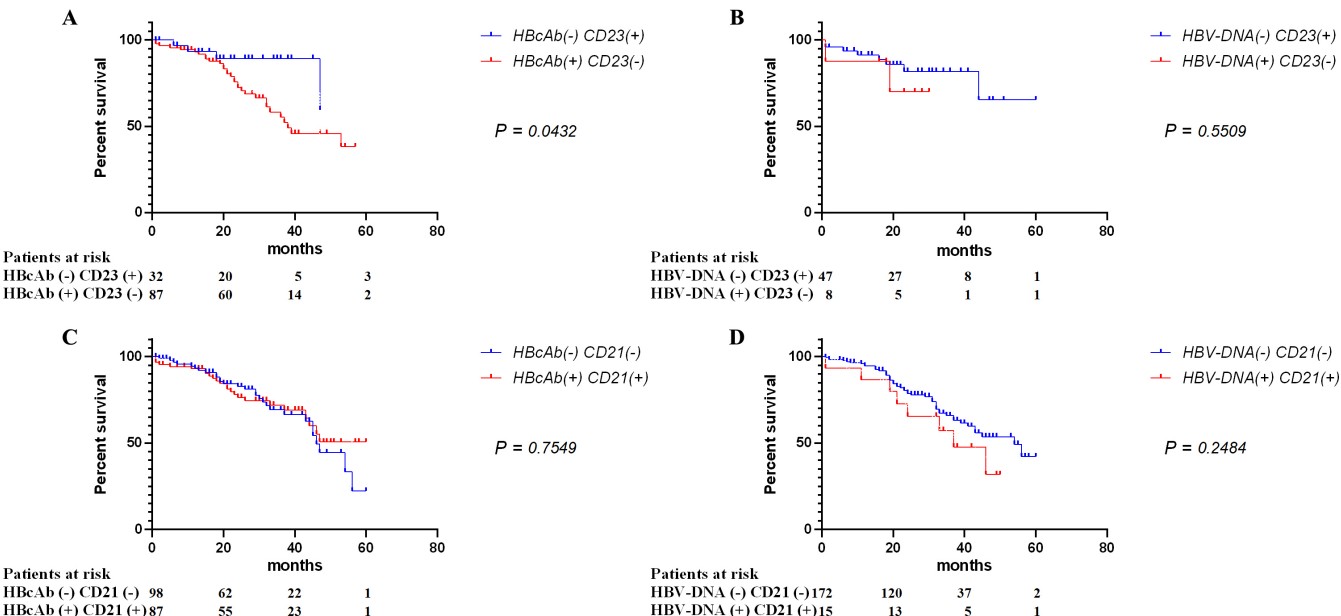

**FIG 3** Five-year OS of DLBCL patients combined with HBV indicators and tissue protein expression. Follow-up patients with DLBCL were grouped according to HBcAb (A) and HBV-DNA (B) combined with tissue CD23 expression, and HBcAb (C) and HBV-DNA (D) combined with tissue CD21 expression. The 5-year OS rates between groups were compared by survival analysis.

in Fig. 1, the prognostic correlation was lost in the grouping of HBV-DNA combined with CD23 (Fig. 3), potentially due to the lack of correlation between HBV-DNA and CD23. Similar results were also obtained in the OS analysis (Fig. S3). Therefore, CD23 can be combined with the HBcAb indicator for the prognostic evaluation of HBV-associated DLBCL.

Notably, unlike CD23, the combination of HBcAb and pathological markers that lack correlation such as CD21 (r = 0.003, *P* = 0.900) does not enhance the prognostic value of CD21, and the indeterminate prognosis associated with CD21 remains unchanged (Fig. 3).

However, due to the censoring of follow-up data, no statistically significant difference was observed in the OS of the combination of qualitative HBcAb and CD23 (Fig. S3). It is necessary to incorporate the quantitative analysis of HBcAb for further validation. Consistent with the results of the qualitative analysis, HBcAb concentrations were significantly reduced in CD23-positive patients (Fig. 4), whereas the HBV-DNA concentration showed no difference between the CD23-negative and -positive groups (Fig. S4). Therefore, we reconfirmed the negative correlation between HBcAb and CD23 from the perspective of quantitative analysis.

In the survival analysis involving HBcAb concentration, DLBCL patients were divided into two groups according to the median concentration of HBcAb combined with CD23 status, and the 5-year OS rate between the groups was significantly different (*P* = 0.048) (Fig. 4B). Consistent results were also obtained in the OS analysis (Fig. S4). Notably, a significant difference in OS between the combination of quantitative HBcAb and CD23 was revealed.

## DISCUSSION

In this study, we found that HBcAb, which was negatively correlated with CD23, had significantly reduced OS rates in DLBCL patients (Table 1; Fig. 3 and 4). To date, several viruses have been implicated in the tumorigenesis and progression of DLBCL, such as Epstein-Barr virus (EBV), hepatitis C virus (HCV), and human immunodeficiency virus (HIV) (25–28). However, unlike EBV, which is often directly oncogenic through

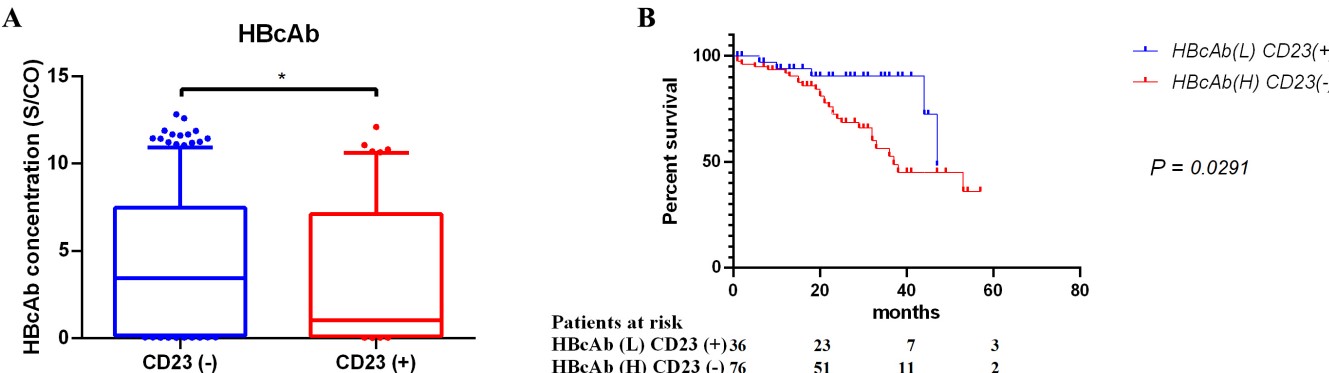

**FIG 4** Correlation and 5-year OS of DLBCL patients in combination with HBcAb concentration and tissue CD23 expression. The correlation between HBcAb concentration and CD23 in DLBCL patients was compared by the Mann-Whitney method, which was illustrated by the box of the median with quartile and whiskers of 2.5–97.5 percentile (A). Patients were classified according to the median concentration of HBcAb and CD23 status, and the 5-year OS rates among groups were compared through survival analysis (B).

mechanisms like latent membrane protein (LMP) expression (29), HBV appears to contribute to DLBCL pathogenesis through viral integration and immune dysregulation. In addition, studies have linked HBV infection with increased cancer risk beyond HCC, including DLBCL. These findings emphasize the importance of HBV management and provide therapeutic insights for DLBCL patients with HBV infection (30).

Our findings indicate the prognostic value of HBcAb as an independent marker in DLBCL, underscoring its importance in the assessment of HBV infection (Fig. 1). One possible reason can be attributed to the presence of *HBx* integration in DLBCL tissues (13, 31). HBxAg, in particular, has been implicated in the promotion of viral replication and the modulation of host cellular processes, potentially contributing to oncogenesis (32). It has been discovered that HBV genome integration events occur more frequently at the 3' end of the *HBx* gene and the 5' end of the Core (*HBc*) gene in HBV-related HCC (33). When the expression of HBxAg increased, the level of HBcAg mostly decreased (34). Given that traditional HBV serological indicators typically lack the detection of trace amounts of HBx protein, therefore, serological testing for HBcAb not only serves as a marker for HBV infection but also may become a key indicator reflecting the presence of HBx, which is crucial for understanding the tumorigenesis associated with HBV. HBcAb is the only independent predictor for virological and serological responses in some cases (35). Regarding other HBV antibodies, the presence of HBsAb is influenced by planned immunization, and it was found that HBsAb was not related to the OS of patients with DLBCL.

Furthermore, a negative correlation between HBcAb and the expression of CD23 in tumor tissues suggests a potential role of CD23 in HBV-associated DLBCL (Table 1; Fig. 4). CD23 was also associated with poor prognosis in follicular lymphomas, which is consistent with our findings in DLBCL (36). As an Fc receptor, CD23 is implicated in the modulation of viral infections (37), and its mechanistic interaction with HBV warrants further investigation. The prognostic value of CD23 as a solitary marker was not evident in our study (Fig. 2), possibly due to limitations in sample size, and requires further exploration. The correlation between CD23 and the survival prognosis of HBV-associated DLBCL may be related to changes in cellular survival and drug resistance levels. As a cell surface molecule, expression changes of CD23 could affect the activation of B cells in the absence of IgE immune complex and the sensitivity of tumor cells to immune checkpoint inhibitors (38, 39), thereby influencing patient prognosis in HBV-associated DLBCL. This result highlights the need to consider the potential role of CD23 in the treatment of HBV-associated DLBCL and how it interacts with HBV infection to jointly affect disease progression and treatment response.

The identified relationship between HBcAb and CD23 in our study complements existing literature (40). Our results suggest that the interplay between HBcAb and CD23 could be pivotal in the progression and prognosis of HBV-associated DLBCL. However, our study reveals that the Bcl-2 protein is closely related to the survival prognosis of DLBCL, whereas its expression is not correlated with serological indicators of HBV status such as HBcAb (Fig. 2; Table 1). HBV infection does not elevate Bcl-2 levels, potentially diminishing its prognostic significance when combined with HBcAb. This result suggests that HBV-associated DLBCL may involve Bcl-2-independent apoptotic pathways. Simultaneously, already low expressed P53 in DLBCL does not exhibit prognostic relevance in the context of HBV infection (Fig. 2; Table 1). In contrast, the combination of HBcAb with CD23 enhances the prognostic utility of CD23, either by amplifying its predictive power or by preserving the prognostic impact of HBcAb (Fig. 3 and 4). This finding is significant for exploring the molecular mechanisms of DLBCL and developing new therapeutic strategies, indicating the need to identify new molecular targets to intervene in the process of HBV-associated DLBCL. Notably, CD21 showed no prognostic value either alone or in combination with other markers (Fig. 2 and 3), contrasting with previous reports (40). This discrepancy may relate to the nature of DNA or antigen-based indicators, and our $P$-values falling within the 0.1–0.3 range (Table S1) indicate a need for multicenter studies to provide more robust evidence.

The distribution of HBV genotypes varies across different regions within China (41), and intergenotypic recombinant HBV strains have also been identified (42). However, it should be noted that the impact of HBV genotypes on DLBCL remains to be elucidated, as no relevant studies have been conducted thus far. Our center lacks experimental data on HBV genotyping, which may limit the generalizability of our findings. It is well-documented that antiviral therapy can significantly influence HBV reactivation rates and clinical outcomes in patients with HBV-associated malignancies (43). Different from the results of other studies (40), our study showed that there was no difference in the prognosis of HBsAg negative and positive patients, which may be affected by conventional antiviral treatment for all patients with HBV reactivation during chemotherapy in our center.

In summary, HBcAb and CD23, as key biomarkers in HBV-associated DLBCL, highlight the assessment of disease prognosis and the formulation of treatment strategies. Future research needs to further clarify their specific mechanisms in HBV-associated DLBCL and how they interact with other molecular pathways, thereby providing a theoretical basis for the development of effective therapeutic approaches. The identification of serum HBcAb and CD23 as prognostic markers for HBV-associated DLBCL may guide clinicians in stratifying patients for risk assessment and treatment planning. These biomarkers could potentially be integrated into existing prognostic models to improve the accuracy of predicting patient outcomes. A prognostic model combining HBcAb and CD23 needs to be developed, validated, and transformed into clinical detection and prognostic evaluation. In addition, we are studying the mechanism of HBV-related CD23, which will help discover HBV-related specific molecules and facilitate the development and clinical translation of potential therapeutic targets.

## ACKNOWLEDGMENTS

We gratefully acknowledge Changming Zhou from the Department of Cancer Prevention, Fudan University Shanghai Cancer Center for providing clinical follow-up data and professional analysis support.

This work was supported by the National Natural Science Foundation of China (grant Nos. 81800190, 82072876, and 82373383).

Yi Rong and Yanchun Wang designed and performed the experiments. Min Wang collected the clinical data. Yaqiong Ma, Yuanchen Liang, and Lvyin Ye analyzed the data. Yi Rong drafted the manuscript. Lin Guo and Renquan Lu were responsible for technical support and participated in discussions. Yanchun Wang edited and supervised the manuscript.

## AUTHOR AFFILIATIONS

[1]Department of Clinical Laboratory, Fudan University Shanghai Cancer Center, Shanghai, China

[2]Department of Oncology, Shanghai Medical College, Fudan University, Shanghai, China

## AUTHOR ORCIDs

Yi Rong  http://orcid.org/0009-0002-3436-4817
Lin Guo  http://orcid.org/0000-0002-8419-3350
Renquan Lu  http://orcid.org/0000-0003-3291-5742
Yanchun Wang  http://orcid.org/0000-0001-8429-9182

## FUNDING

| Funder | Grant(s) | Author(s) |
| --- | --- | --- |
| MOST \| National Natural Science Foundation of China (NSFC) | 81800190 | Yanchun Wang |
| MOST \| National Natural Science Foundation of China (NSFC) | 82072876,82373383 | Renquan Lu |

## AUTHOR CONTRIBUTIONS

Yi Rong, Methodology, Writing – original draft | Ming Wang, Data curation | Yaqiong Ma, Formal analysis | Yuanchen Liang, Formal analysis | Lvyin Ye, Formal analysis | Lin Guo, Resources, Supervision | Renquan Lu, Resources, Supervision | Yanchun Wang, Methodology, Supervision, Writing – original draft, Writing – review and editing

## DATA AVAILABILITY

The datasets used and/or analyzed during the current study are available from the corresponding author upon reasonable request.

## ETHICS APPROVAL

Medical history data collection in this study were approved by the ethics committee of Fudan University Shanghai Cancer Center (050432-4-2307E).

## ADDITIONAL FILES

The following material is available online.

### Supplemental Material

**Supplemental figures (Spectrum03170-24-s0001.docx).** Fig. S1 to S4.
**Table S1 (Spectrum03170-24-s0002.docx).** Correlation between tissue protein expression and all HBV indicators detected in DLBCL.

### Open Peer Review

**PEER REVIEW HISTORY (review-history.pdf).** An accounting of the reviewer comments and feedback.

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
