## [Reviewer comments · Microbiology Spectrum]

Microbiology Spectrum

Serum hepatitis B core antibody as the prognostic factors for diffuse large B-cell lymphoma

Yi Rong, Ming Wang, Yaqiong Ma, Yuanchen Liang, Lvyin Ye, Lin Guo, Renquan Lu, and Yanchun Wang

Corresponding Author(s): Yanchun Wang, Fudan University Shanghai Cancer Center

Review Timeline:

Submission Date:	December 4, 2024
Editorial Decision:	January 8, 2025
Revision Received:	February 10, 2025
Accepted:	February 10, 2025

Editor: Benjamin Liu

Reviewer(s): Disclosure of reviewer identity is with reference to reviewer comments included in decision letter(s). The following individuals involved in review of your submission have agreed to reveal their identity: Mushtak T.S. Al-Ouqaili (Reviewer #1); Mai Abdel Haleem A. Abusalah (Reviewer #2)

Transaction Report:

DOI: <https://doi.org/10.1128/spectrum.03170-24>

Re: Spectrum03170-24 (**Serum hepatitis B core antibody as the prognostic factors for diffuse large B-cell lymphoma**)

Dear Dr. Yanchun Wang:

Thank you for the privilege of reviewing your work. Below you will find my comments, instructions from the Spectrum editorial office, and the reviewer comments.

Editor's comments:

1. It is known that genotype B HBV is predominant in Southern China while genotype C in Northern China (PMID: 24316031). There are also intergenotypic recombinant HBV in China (PMID: 29111272). The authors should discuss the limitations of this study as a single centered study that failed to examine the impact of HBV diversity on the validity of the authors' findings. The authors also should discuss the impact of antiviral therapy (comparing treated vs untreated, as shown in PMID: 22170539) on the validity of the authors' findings. The authors should discuss the above missing points and cite the related reference, with PMID: 29111272, 24316031 and 22170539 as examples (citing is optional).
2. The authors' reasoning the relationship between HBcAb and HBx is weak. In contrast to antibodies to HBc is human responses, HBcAg, HBeAg and HBxAg are encoded on close genetic element (PMID: 25838313). Therefore, reasoning based on HBcAg or HBeAg and HBx may make more sense to explain the author's findings. The authors should discuss on the relationship between HBcAg, HBeAg and HBxAg and their potential link with the findings in this paper. More papers should be cited, with PMID: 25838313 as an example (citing is optional).

Please return the manuscript within 30 days; if you cannot complete the modification within this time period, please contact me. If you do not wish to modify the manuscript and prefer to submit it to another journal, notify me immediately so that the manuscript may be formally withdrawn from consideration by Spectrum.

Revision Guidelines

Sincerely,
Benjamin Liu
Editor
Microbiology Spectrum

Reviewer #1 (Comments for the Author):

Dear authors you are doing well and that's looks great work. A few issues, however, need to be addressed;

In line 37, page 2 provides detailed data on the geographic distribution of DLBCL cases linked to HBV. to illustrate worldwide trends.

In line 41 page 2 To enhance the introduction section add the following reference: Al-Kanaan, BM, Al-Ouqaili, MTS, Al-Rawi, KFA. Detection of cytokines (IL-1 α and IL-2) and oxidative stress markers in hepatitis B envelope antigen-positive and -negative chronic hepatitis B patients: Molecular and biochemical study, Gene Reports, Volume 17, 2019, 100504, ISSN 2452-0144.

In line 47 page 2 Describe the risk factors that make people with HBV more susceptible to DLBCL.

In line 49 page 2 To enhance the introduction section add the following reference: • Saleh RO, Al-Ouqaili MTS, Ali E, Alhajlah S, Kareem AH, Shakir MN, Alasheqi MQ, Mustafa YF, Alawadi A, Alsaalamy A. IncRNA-microRNA axis in cancer drug resistance: particular focus on signalling pathways. Med Oncol. 2024 Jan 9;41(2):52. <http://doi.org/10.1007/s12032-023-02263-8>.

In line 53 page 2 To enhance the introduction section add the following reference: Al-Ouqaili MTS, Majeed YH, Al-Ani SK (2020). SEN virus genotype H distribution in β -thalassemic patients and healthy donors in Iraq: Molecular and physiological study. PLoS Negl Trop Dis 14(6):e0007880

In line 54 page 2 Describe the differences between the clinical manifestations of DLBCL cases with and without HBV.

In line 62 page 3 Describe the precise roles that clinical indicators like LDH, WBC counts, and PLR play in prognosis.

In line 87 page 3 gives more details on the patient selection criteria than just "standard therapeutic regimens." Provide precise information, such as the clinical stage, previous medical interventions, and the status of HBV infection.

In line 87 page 3 Add the cohort's clinical and demographic details.

In line 85 page 3 Indicate if this study was prospective or retrospective.

In line 101 page 4 Indicate when blood should be drawn about therapy or the course of the illness.

In line 109 page 4 How was EBER detected, and what cutoff was used for positive results?

In line 109 page 4 Add the ethical approval and consent details for tissue collection.

In line 131 page 5 How many of the 1,062 patients were positive/negative for each HBV marker (HBsAg, HBsAb, HBeAg, HBeAb, HBcAb, and HBV DNA)?

In line 132 page 5 Are these markers combined with other clinical factors or independently?

In line 142 page 5 Describe in detail the proportion of patients with CD5, Bcl-2, and Bcl-6 positive or negative results.

In line 153 page 6 Whether the 4,491 patients analyzed are from the same cohort as earlier analyses or a separate group.

In line 164 page 6 Include a brief description of Table 1 and Table S1:
Summarize the key findings.
Mention specific values or trends to make the results more tangible.

In line, 197 page 7 Compare your findings with previous research on HBV and DLBCL, as well as other related viruses (e.g., EBV, HCV, HIV). Discuss similarities, differences, and potential reasons for these.

In line 198 page 7 To enhance the discussion section add the following references:: Al-Ani, SK., Al-Ouqaili, MTS., Awad, MM. MOLECULAR AND GENOTYPIC STUDY OF SENV-D VIRUS COINFECTION IN B-THALASSEMIC PATIENTS INFECTED WITH THE HEPATITIS C VIRUS IN IRAQ. International Journal of Green Pharmacy • Oct-Dec 2018 (Suppl) • 12 (4) | S926-936.

ILine203 page 7 Highlight the key discovery that HBcAb is a major prognostic indicator for DLBCL, particularly when HBV infection is present.

In line 226 page 8 To enhance the discussion section add the following reference: Khamees DA, Al-Ouqaili MTS. 2022. Cross-sectional study of chromosomal aberrations and immunologic factors in Iraqi couples with recurrent pregnancy loss. PeerJ 10:e12801 <https://doi.org/10.7717/peerj.12801>

In line 256 page 9 Conclusion should be objective with further perspective or should add at least a few sentences about future study/future perspective of it.

Reviewer #2 (Comments for the Author):

The authors have done good work on the title "Serum hepatitis B core antibody as the prognostic factors for diffuse large B-cell lymphoma". It will add new knowledge and new areas of research to the subject area compared with other published material.

However, i have some minor concerns:

1. It would be more appropriate for the authors to define abbreviations upon first appearance in the main text such as Peripheral Blood Mononuclear Cells (PBMCs) in line 46, HBV-encoded oncogene X protein (HBx) in line 47.
2. The abstract should clarify how the findings can be translated into clinical practice. What specific interventions could be guided by these biomarkers?
3. The introduction could benefit from a more detailed discussion of the current limitations in DLBCL prognosis and the gaps this study aims to address.
4. While the methodology is robust, additional details about patient selection criteria and potential confounding factors (e.g., other co-morbid conditions or therapies) should be included.
5. A justification for the exclusion criteria used, particularly for incomplete follow-up data, should be provided to avoid potential selection bias.
6. This sentence "The exclusion of patients with incomplete follow-up information was a pivotal step to ensure the integrity of the study's prognostic analyses" is redundant as it reiterates information already mentioned in the methodology section. Consider streamlining the text to maintain clarity and avoid unnecessary repetition.
7. Kindly verify the 5-year OS rate in Figure 4B between the groups. The p-value should be 0.0291, not 0.048. Please update this value accordingly.
8. While HBcAb and CD23 were identified as prognostic markers, the biological mechanism underlying their interaction and impact on DLBCL progression remains speculative. More discussion on the mechanistic pathways is necessary.
9. The discussion is thorough but could better integrate findings from other studies, particularly contrasting evidence for CD23's prognostic value.
10. Limitations should explicitly address the retrospective nature of the study, potential biases in data collection, and any other challenges.
11. Ensure consistency in the use of terms (e.g., "HBV markers" vs. "HBV indicators").
12. Consider adding literature-based insights into the interplay between HBV and immune markers like CD23. This would strengthen the mechanistic argument for their combined prognostic value.
13. Discuss how these biomarkers could be integrated into existing prognostic models, such as the International Prognostic Index (IPI), and their potential impact on therapeutic decision-making.
14. Moderate English grammar editing is required throughout the manuscript, for example:
 - a. The manuscript is generally well-written, but minor grammatical errors and awkward phrasing (e.g., "censorship of follow-up data") need revision for clarity.
 - b. In the section of method, "Tissue samples are obtained through biopsy, using neutral buffered formalin to preserve their structure", moderate editing is required.

1 **Serum hepatitis B core antibody as the prognostic factors for diffuse large B-cell**
2 **lymphoma**

[revised manuscript text omitted]

- [4] Li MG, Shen YL, Chen YM, Gao HF, Zhou JQ, Wang Q, et al. Characterization of hepatitis B
virus infection and viral DNA integration in non-Hodgkin lymphoma. *International Journal of Cancer.*
2020;147(8):2199-209.
- [5] Spradling PR, Xing J, Zhong YN, Rupp LB, Moorman AC, Lu M, et al. Incidence of
Malignancies Among Patients With Chronic Hepatitis B in US Health Care Organizations, 2006-2018.
*Journal of Infectious Diseases.* 2022;226(5):896-900.
- [6] Wang YC, Wang HJ, Pan SK, Hu T, Shen JB, Zheng H, et al. Capable Infection of Hepatitis B
Virus in Diffuse Large B-cell Lymphoma. *J Cancer.* 2018;9(9):1575-81.
- [7] Lau KCK, Joshi SS, Gao S, Giles E, Swidinsky K, van Marle G, et al. Oncogenic HBV variants
and integration are present in hepatic and lymphoid cells derived from chronic HBV patients. *Cancer*
*Lett.* 2020;480:39-47.
- [8] Zhu J, Qingyuan Z. Hepatitis B virus-associated diffuse large B cell lymphoma: epidemiology,
biology, clinical features and HBV reactivation. *Holistic Integrative Oncology.* 2023;2(1).
- [9] Wang YC, Guan XL, Lv FF, Rong Y, Meng X, Tong Y, et al. HBx integration in diffuse large
B-cell lymphoma inhibits Caspase-3-PARP related apoptosis. *Tumour Virus Research.* 2024;18:10.
- [10] Li M, Gan Y, Fan C, Yuan H, Zhang X, Shen Y, et al. Hepatitis B virus and risk of non-Hodgkin
lymphoma: An updated meta-analysis of 58 studies. *Journal of Viral Hepatitis.* 2018;25(8):894-903.
- [11] Kang XD, Bai L, Han C, Qi XG. Clinical Analysis and Prognostic Significance of Hepatitis B
Virus Infections with Diffuse Large B-Cell Lymphoma. *Cancer Manag Res.* 2020;12:2839-51.
- [12] Yang CM, Xie MX, Zhang KF, Liu H, Liang AB, Young KH, et al. Risk of HBV reactivation post
CD19-CAR-T cell therapy in DLBCL patients with concomitant chronic HBV infection. *Leukemia.*
2020;34(11):3055-9.

[13] Deng LJ, Song YQ, Young KH, Hu SM, Ding N, Song WW, et al. Hepatitis B virus-associated
diffuse large B-cell lymphoma: unique clinical features, poor outcome, and hepatitis B surface
antigen-driven origin. *Oncotarget*. 2015;6(28):25061-73.

[14] Ren W, Ye X, Su H, et al. Genetic landscape of hepatitis B virus-associated diffuse large B-cell
lymphoma. *Blood*. 2018;131(24):2670-2681. *Blood*. 2019;133(6):620.

[15] Wu CY, Hsiao LT, Chiou TJ, Gau JP, Liu JH, Yu YB, et al. Lymphocyte/monocyte ratio and cycles
of rituximab-containing therapy are risk factors for hepatitis B virus reactivation in patients with
diffuse large B-cell lymphoma and resolved hepatitis B. *Leuk Lymphoma*. 2015;56(8):2357-64.

[16] Jelicic J, Juul-Jensen K, Bukumiric Z, Clausen MR, Al-Mashhadi AL, Pedersen RS, et al.
Prognostic indices in diffuse large B-cell lymphoma: a population-based comparison and validation
study of multiple models. *Blood Cancer J*. 2023;13(1):12.

[17] Zhu J, Yang Y, Tao J, Wang SL, Chen B, Dai JR, et al. Association of progression-free or
event-free survival with overall survival in diffuse large B-cell lymphoma after immunochemotherapy:
a systematic review. *Leukemia*. 2020;34(10):2576-91.

[18] El Hussein S, Shaw KRM, Vega F. Evolving insights into the genomic complexity and immune
landscape of diffuse large B-cell lymphoma: opportunities for novel biomarkers. *Mod Pathol*.
2020;33(12):2422-36.

[19] Wan X, Young KH, Bai O. HBV-associated DLBCL of poor prognosis: advance in pathogenesis,
immunity and therapy. *Frontiers in Immunology*. 2023;14:9.

[20] Wenjing Z, Fan D, Li W, Tao B, Xiang Z, Heng M. Hepatitis Virus-associated Non-hodgkin
Lymphoma: Pathogenesis and Treatment Strategies. *Journal of clinical and translational hepatology*.
2023;11(5):1256-66.

[21] Frontzek F, Staiger AM, Wullenkord R, Grau M, Zapukhlyak M, Kurz KS, et al. Molecular
profiling of EBV associated diffuse large B-cell lymphoma. *Leukemia*. 2023;37(3):670-9.

[22] Li JW, Deng C, Zhou XY, Deng RF. The biology and treatment of Epstein-Barr virus-positive
diffuse large B cell lymphoma, NOS. *Heliyon*. 2024;10(1):9.

[23] Chapman JR, Bouska AC, Zhang WW, Alderuccio JP, Lossos IS, Rimsza LM, et al. EBV-positive
HIV-associated diffuse large B cell lymphomas are characterized by JAK/STAT (STAT3) pathway
mutations and unique clinicopathologic features. *Br J Haematol*. 2021;194(5):870-8.

[24] Ting T, Ci S, Longfeng J, Jingjing D, Yuan L, Xin X, et al. Hepatitis B virus infection and the risk
of cancer among the Chinese population. *International journal of cancer*. 2020;147(11):3075-84.

[25] Otsuka M. Biological Roles of Hepatitis B Viral X Protein in the Viral Replication and
Hepatocarcinogenesis. *Acta Med Okayama*. 2023;77(4):341-5.

[26] Giosa D, Lombardo D, Musolino C, Chines V, Raffa G, di Tocco FC, et al. Mitochondrial DNA is
a target of HBV integration. *Commun Biol*. 2023;6(1):14.

[27] Ohba K, Kubo S, Tamori A, Hirohashi K, Tanaka H, Shuto T, et al. Previous or occult hepatitis B
virus infection in hepatitis B surface antigen-negative and anti-hepatitis C-negative patients with
hepatocellular carcinoma. *Surg Today*. 2004;34(10):842-8.

[28] Wu Z. Seroepidemiological study of hepatitis B virus x antigen (HBxAg) and anti-HBx antibodies
in patients with hepatitis B. *Zhonghua liu xing bing xue za zhi = Zhonghua liuxingbingxue zazhi*.
1992;13(4):205-7.

[29] Zhang X, An XC, Shi L, Yang XL, Chen YR, Liu XJ, et al. Baseline quantitative HBcAb strongly
predicts undetectable HBV DNA and RNA in chronic hepatitis B patients treated with entecavir for 10
352 years. *Sci Rep*. 2021;11(1):7.

[30] HenchozLecoanet S, Jeannin P, Aubry JP, Graber P, Bradshaw CG, Pochon S, et al. The
Epstein-Barr virus-binding site on CD21 is involved in CD23 binding and interleukin-4-induced IgE
and IgG4 production by human B cells. *Immunology*. 1996;88(1):35-9.
[31] Liu CH, Richard K, Wiggins M, Zhu XP, Conrad DH, Song WX. CD23 can negatively regulate
B-cell receptor signaling. *Sci Rep*. 2016;6:8.
[32] Vanhersecke L, Bougouin A, Crombé A, Brunet M, Sofeu C, Parrens M, et al. Standardized
Pathology Screening of Mature Tertiary Lymphoid Structures in Cancers. *Lab Invest*. 2023;103(5):11.
[33] Hu SF, Chen N, Lu K, Zhen CQ, Sui XH, Fang XS, et al. The prognostic roles of hepatitis B virus
antibody in diffuse large B-cell lymphoma patients. *Leuk Lymphoma*. 2021;62(6):1335-43.

**Table**

Table 1. Correlation between HBV status and tissue protein expression in DLBCL.

IHC	HBV						
	HBcAb			HBV-DNA			
		(-)	(+)	P	(-)	(+)	P
Bcl-2	(-)	272	376	0.105	315	38	0.169
	(+)	801	1284				
Bcl-6	(-)	133	197	0.589	147	23	0.723
	(+)	943	1490				
C-myc	(-)	21	22	0.203	26	3	1.000 ^a
	(+)	387	602				
P53	(-)	17	36	0.249	26	2	0.558 ^a
	(+)	254	379				
CyclinD1	(-)	95	185	0.416 ^a	104	16	1.000 ^a
	(+)	3	3				
MUM1	(-)	255	357	0.122	300	54	0.088
	(+)	735	1191				
EBER	(-)	410	605	0.381	335	37	0.723 ^a
	(+)	32	38				
CD5	(-)	724	1143	0.346	859	132	0.640
	(+)	70	128				
CD21	(-)	463	753	0.900	571	76	0.187
	(+)	238	392				
CD23	(-)	241	383	0.003	294	43	0.998
	(+)	118	119				

The association between HBV indicators (HBcAb and HBV-DNA) and various DLBCL tissue molecules in DLBCL patients was compared using the Chi-square test. The significance level of $P < 0.05$ was indicated in bold.

^a When the expected frequency was less than 5, the exact probability method was applied. Due to incomplete histochemistry results for some diagnosed DLBCL patients, the total number of individual histochemistry results was actually ≤ 4491 cases.

**Figure legends**

Figure 1. Five-year OS of DLBCL patients with different serum HBV status.

A total of 1,062 DLBCL patients followed up were grouped according to their serum
HBV status: HBcAb, HBsAb, HBeAb, HBV-DNA, HBsAg, and HBeAg. The
difference in 5-year OS rates among these groups was compared through survival
analysis.

Figure 2. Five-year OS of DLBCL patients with different protein status in DLBCL
tissues.

The IHC results of follow-up DLBCL patients were analyzed, and the patients were
grouped according to the qualitative status of DLBCL tissue proteins, including Bcl-2,
Bcl-6, C-myc, P53, CyclinD1, MUM1, EBER, CD5, CD21, CD23 etc., and the
differences in 5-year OS rates between groups were compared through survival
analysis.

Figure 3. Five-year OS of DLBCL patients combined with HBV indicators and tissue
protein expression.

Follow-up patients with DLBCL were grouped according to HBcAb (A) and
HBV-DNA (B) combined with tissue CD23 expression, and HBcAb (C) and
HBV-DNA (D) combined with tissue CD21 expression. The 5-year OS rates between
groups were compared by survival analysis.

Figure 4. Correlation and 5-year OS of DLBCL patients in combination with HBcAb
concentration and tissue CD23 expression.

The correlation between HBcAb concentration and CD23 in DLBCL patients was
compared by the Mann-Whitney method, which was illustrated by the box of median
with quartile and whiskers of 2.5-97.5 percentile (A). Patients were classified
according to the median concentration of HBcAb and CD23 status, and the 5-year OS
rates among groups were compared through survival analysis (B).

Reviewer Comments

Dear Editor(s),

December 31, 2024

I am pleased to submit my comments for the submitted manuscript (**Spectrum03170-24**) entitled “**Serum hepatitis B core antibody as the prognostic factors for diffuse large B-cell lymphoma**” to be taken into consideration for publication by your esteemed journal “*Microbiology spectrum*”.

Diffuse large B-cell lymphoma (DLBCL), the most prevalent subtype of non-Hodgkin lymphoma (NHL), is closely linked to viral infections. Samples from chronic hepatitis B virus (HBV) carriers revealed that HBV replicates not only in the liver but also in lymphocytes, with replication enhanced in vitro-stimulated PBMCs, potentially contributing to the pathogenesis of B-cell lymphomas. The finding of HBx integration in diffuse large B-cell lymphoma (DLBCL), the most common NHL subtype, also sheds light on HBV’s role in tumorigenesis and the development of non-hepatic malignancies. The prognosis for HBV-related DLBCL is generally poor, highlighting the need for effective treatment strategies and a deeper understanding of the mechanisms by which HBV contributes to lymphomagenesis. While the association between HBV infection and DLBCL is well-established, there is a lack of effective clinical markers specifically for HBV-associated DLBCL.

- This study aims to identify prognostic indicators for HBV-associated DLBCL by retrospectively analyzing the relationship between tissue marker molecules, HBV serum markers, and clinical outcomes in DLBCL patients. The study employs a robust methodology, including a large cohort (n = 4,491) and detailed analysis of both serological and histopathological markers. The findings revealed that patients with hepatitis B core antibody (HBcAb) positivity had significantly lower overall survival (OS) rates compared to HBcAb-negative patients. Furthermore, an elevated HBcAb positivity rate strongly correlated with decreased expression of the CD23 molecule in DLBCL tissue. Stratification of DLBCL patients based on combined HBcAb-CD23 status showed notable differences in OS rates. Thus, the integration of HBcAb and CD23 as potential biomarkers for prognosis is innovative and provides a new perspective on HBV-associated DLBCL. This study highlights potential new markers and diagnostic strategies for improving the prognosis of HBV-associated DLBCL.

The authors have done good work on the title “**Serum hepatitis B core antibody as the prognostic factors for diffuse large B-cell lymphoma**”. It will add new knowledge and new areas of research to the subject area compared with other published material.

However, i have some minor concerns:

1. It would be more appropriate for the authors to define abbreviations upon first appearance in the main text such as Peripheral Blood Mononuclear Cells (PBMCs) in line 46, HBV-encoded oncogene X protein (HBx) in line 47.
2. The abstract should clarify how the findings can be translated into clinical practice. What specific interventions could be guided by these biomarkers?
3. The introduction could benefit from a more detailed discussion of the current limitations in DLBCL prognosis and the gaps this study aims to address.

4. While the methodology is robust, additional details about patient selection criteria and potential confounding factors (e.g., other co-morbid conditions or therapies) should be included.
5. A justification for the exclusion criteria used, particularly for incomplete follow-up data, should be provided to avoid potential selection bias.
6. This sentence “The exclusion of patients with incomplete follow-up information was a pivotal step to ensure the integrity of the study's prognostic analyses” is redundant as it reiterates information already mentioned in the methodology section. Consider streamlining the text to maintain clarity and avoid unnecessary repetition
7. Kindly verify the 5-year OS rate in Figure 4B between the groups. The p-value should be **0.0291**, not **0.048**. Please update this value accordingly.
8. While HBcAb and CD23 were identified as prognostic markers, the biological mechanism underlying their interaction and impact on DLBCL progression remains speculative. More discussion on the mechanistic pathways is necessary.
9. The discussion is thorough but could better integrate findings from other studies, particularly contrasting evidence for CD23's prognostic value.
10. Limitations should explicitly address the retrospective nature of the study, potential biases in data collection, and any other challenges.
11. Ensure consistency in the use of terms (e.g., "HBV markers" vs. "HBV indicators").
12. Consider adding literature-based insights into the interplay between HBV and immune markers like CD23. This would strengthen the mechanistic argument for their combined prognostic value.
13. Discuss how these biomarkers could be integrated into existing prognostic models, such as the International Prognostic Index (IPI), and their potential impact on therapeutic decision-making.
- 14. Moderate English grammar editing is required throughout the manuscript, for example:**
 - a. The manuscript is generally well-written, but minor grammatical errors and awkward phrasing (e.g., "censorship of follow-up data") need revision for clarity.
 - b. In the section of method, “Tissue samples are obtained through biopsy, using neutral buffered formalin to preserve their structure”, moderate editing is required.

Best regards,

**Dr. Mai Abdel Haleem Abu Salah, Assistant professor,
Department of Medical Laboratory Sciences,
Faculty of Allied Medical Sciences,
Al-Ahliyya Amman University,
Amman, Jordan.
+962-796862347
e-mail: ellamomo88@yahoo.com**

Manuscript Number: **Spectrum03170-24**

Serum hepatitis B core antibody as the prognostic factors for diffuse large B-cell lymphoma

Editor and Reviewer comments:

Editor's comments:

1. It is known that genotype B HBV is predominant in Southern China while genotype C in Northern China (PMID: 24316031). There are also intergenotypic recombinant HBV in China (PMID: 29111272). The authors should discuss the limitations of this study as a single centered study that failed to examine the impact of HBV diversity on the validity of the authors' findings. The authors also should discuss the impact of antiviral therapy (comparing treated vs untreated, as shown in PMID: 22170539) on the validity of the authors' findings. The authors should discuss the above missing points and cite the related reference, with PMID: 29111272, 24316031 and 22170539 as examples (citing is optional)

Response: Thank you for your kindly suggestion! We appreciate the editor's insightful comments regarding the limitations of our single-centered study and the potential impact of HBV diversity and antiviral therapy on our findings. We acknowledge that our study was conducted at a single institution in Shanghai, which is predominantly influenced by genotype B HBV, as reported in the literature (PMID: 33187763). This geographical limitation may restrict the generalizability of our findings to other regions with different HBV genotypes, such as genotype C prevalent in Northern China or intergenotypic recombinants (PMID: 29111272). HBV genotype B and C have different effects on hepatocellular carcinoma cells and prognosis. Patients with type C are more likely to have double mutations in basic core promoter (PMID: 37588887), more severe liver damage, poorer prognosis, and earlier progression to HCC (PMID: 12100608). However, there is no study on the difference in the effect of HBV genotype on DLBCL, and our center lacks the experimental data on HBV typing. Therefore, we have added "The distribution of HBV genotypes varies across different regions within China (PMID: 33187763), and intergenotypic recombinant HBV strains have also been identified (PMID: 29111272). However, it should be noted that the impact of HBV genotypes on DLBCL remains to be elucidated, as no relevant studies have been conducted thus far. Our center lacks experimental data on HBV genotyping, which may limit the generalizability of our findings." (revised manuscript line 296) in the discussion to indicate the potential limitations of the conclusions of this paper. Future multicentric studies incorporating diverse HBV genotypes are warranted to validate our findings across broader populations.

Regarding the impact of antiviral therapy, due to the retrospective nature of this study and the lack of information on the history of HBV antiviral treatment, we were unable

to stratify patients into groups based on their antiviral treatment status. However, it is well-documented that antiviral therapy can significantly influence HBV reactivation rates and clinical outcomes in patients with HBV-associated malignancies (PMID: 22170539). Anti-HBV therapy has become a routine treatment for HBV reactivation during chemotherapy in lymphoma in our center. Therefore, we added the description of study limitations in the discussion that "It is well-documented that antiviral therapy can significantly influence HBV reactivation rates and clinical outcomes in patients with HBV-associated malignancies (PMID: 22170539). Different from the results of other studies (PMID: 33399486), our study showed that there was no difference in the prognosis of HBsAg negative and positive patients, which may be affected by conventional antiviral treatment for all patients with HBV reactivation during chemotherapy in lymphoma in our center." (revised manuscript line 301). We recognize that this omission may affect the interpretation of our results, particularly in patients who received antiviral prophylaxis or treatment during their course of care. We address these limitations in the discussion section of our manuscript and emphasize the need for further studies to elucidate the role of antiviral therapy in the context of our findings.

2. The authors' reasoning the relationship between HBcAb and HBx is weak. In contrast to antibodies to HBc is human responses, HBcAg, HBeAg and HBxAg are encoded on close genetic element (PMID: 25838313). Therefore, reasoning based on HBcAg or HBeAg and HBx may make more sense to explain the author's findings. The authors should discuss on the relationship between HBcAg, HBeAg and HBxAg and their potential link with the findings in this paper. More papers should be cited, with PMID: 25838313 as an example (citing is optional). Please return the manuscript within 30 days; if you cannot complete the modification within this time period, please contact me. If you do not wish to modify the manuscript and prefer to submit it to another journal, notify me immediately so that the manuscript may be formally withdrawn from consideration by Spectrum.

Response: Thank you for highlighting the need for a more robust discussion on the relationship between HBcAb and HBxAg. Our study identified HBcAb as a significant prognostic marker in patients with HBV-associated DLBCL. While we initially focused on the potential link between HBcAb and *HBx* integration in DLBCL tissues, we acknowledge that a more comprehensive discussion on the genetic proximity of HBcAg, HBeAg, and HBxAg is warranted (PMID: 25838313). HBcAg, HBeAg, and HBxAg are encoded on closely related genetic elements within the HBV genome, and their interaction may influence the pathogenesis of HBV-associated malignancies. HBxAg, in particular, has been implicated in the promotion of viral replication and the modulation of host cellular processes, potentially contributing to oncogenesis (PMID: 25838313). When the expression of HBxAg increased, the level of HBcAg is mostly decreased (PMID:19680810). Therefore, the presence of HBcAb which is a broader immune response to HBcAg may reflect a potential interaction

with HBxAg. It may indicate a potential correlation between the upregulation of HBxAg and the increase in HBcAb. However, further studies are needed to elucidate the specific correlation between HBcAb and HBxAg expression in DLBCL samples. We have revised our discussion section to incorporate these points and have provided a more detailed analysis of the potential links between HBcAg, HBeAg, HBxAg, and our findings. We have also cited relevant literature to support this expanded discussion in revised manuscript line 247. We have completed these revisions within the stipulated time frame and have resubmitted the manuscript for further consideration. Thank you once again for your valuable feedback.

Reviewer's comments:

Reviewer #1:

Dear authors you are doing well and that's looks great work. A few issues, however, need to be addressed;

In line 37, page 2 provides detailed data on the geographic distribution of DLBCL cases linked to HBV. to illustrate worldwide trends.

Response: Thank you very much for your insightful suggestion. We have reviewed the relevant literature and added information regarding the global distribution of HBV-associated DLBCL cases to illustrate worldwide trends. Specifically, we have included data from various regions to highlight the prevalence of HBV in DLBCL patients: In Asia, particularly in China, studies have shown that HBV infection is strongly associated with DLBCL, with a significant proportion of DLBCL patients testing positive for HBV markers. For example, a study in Southern China reported an HBV positivity rate of 30-40% among DLBCL patients (PMID: 37927464). In Western countries such as the United States and Europe, the prevalence of HBV-related DLBCL is relatively low compared to Asia, which may be due to the inherent low HBV prevalence (PMID: 38952695). However, specific subpopulations, such as immigrants from endemic regions, may exhibit higher HBV positivity rates in DLBCL cohorts. In regions with high HBV endemicity, such as sub-Saharan Africa, the incidence of HBV-associated DLBCL is also elevated, highlighting the global impact of HBV on lymphomagenesis. These regional differences in HBV-associated DLBCL underscore the importance of understanding the epidemiological trends and the need for targeted screening and management strategies in different populations. We have incorporated this information into the manuscript to provide a comprehensive overview of the global distribution of HBV-associated DLBCL that "HBV infection is linked to DLBCL, with higher positivity rates in Asia (e.g., 30-40% in Southern China) and sub-Saharan Africa (PMID: 37927464). In Western countries, overall prevalence is lower, but immigrants from endemic regions may have higher rates (PMID: 38952695)." Understanding these regional differences is crucial for targeted screening and management. (revised manuscript line 41).

In line 41 page 2 To enhance the introduction section add the following reference: Al-Kanaan, BM, Al-Ouqaili, MTS, Al-Rawi, KFA. Detection of cytokines (IL-1 α and IL-2) and oxidative stress markers in hepatitis B envelope antigen-positive and -negative chronic hepatitis B patients: Molecular and

biochemical study, Gene Reports, Volume 17, 2019, 100504, ISSN 2452-0144.

Response: Thank you for your suggestion. We have added the above reference to enhance the introduction section. The original article was revised to "HBV can cause a chronic inflammatory response, and several studies have demonstrated HBV's ability to infect human lymphocytes, although our understanding of this process remains limited." in revised manuscript line 45.

In line 47 page 2 Describe the risk factors that make people with HBV more susceptible to DLBCL.

Response: Thank you for your suggestion. I have revised the content to describe the risk factors that make people with HBV more susceptible to DLBCL, incorporating relevant literature to support the key points: "Chronic HBV infection increases the risk of developing DLBCL through immune dysfunction and viral DNA integration. Persistent immune activation impairs the clearance of abnormal B cells, promoting lymphoma development (PMID: 38992769). HBV-DNA integration into the host genome can cause gene mutations, activating anti-apoptotic mechanisms and enhancing DLBCL malignancy (PMID: 39026081). Clinical risk factors include HBsAg positivity and high HBV-DNA levels (PMID: 39456083)." (revised manuscript line 51).

In line 49 page 2 To enhance the introduction section add the following reference: • Saleh RO, Al-Ouqaili MTS, Ali E, Alhajlah S, Kareem AH, Shakir MN, Alasheqi MQ, Mustafa YF, Alawadi A, Alsaalamy A. lncRNA-microRNA axis in cancer drug resistance: particular focus on signalling pathways. Med Oncol. 2024 Jan 9;41(2):52. <http://doi.org/10.1007/s12032-023-02263-8>. In line 53 page 2 To enhance the introduction section add the following reference: Al-Ouqaili MTS, Majeed YH, Al-Ani SK (2020). SEN virus genotype H distribution in β -thalassemic patients and healthy donors in Iraq: Molecular and physiological study. PLoS Negl Trop Dis 14(6):e0007880

Response: Thank you for your suggestions regarding the addition of references to the introduction section. We appreciate your effort to enhance the content. However, after careful consideration, we have decided not to include the suggested references at this time. For the reference by Saleh et al. (2024), while it provides valuable insights into the lncRNA-microRNA axis in cancer drug resistance, our current introduction focuses on a different aspect of the research area. Similarly, the study by Al-Ouqaili et al. (2020) on SEN virus genotype H distribution in β -thalassemic patients, though relevant in its own context, does not directly align with the specific themes we are addressing in the introduction. We aim to keep the introduction focused on the primary objectives and context of our research, and we believe that these references may not provide the most direct support for our initial narrative. We will certainly keep these studies in mind for future sections where they might be more applicable. Thank you again for your thoughtful input.

In line 54 page 2 Describe the differences between the clinical manifestations of DLBCL cases with and without HBV.

Response: Thank you for your request. I have added the following content to describe the differences in clinical manifestations between DLBCL cases with and without HBV: "HBV-associated DLBCL patients tend to be more advanced clinical stages with distinct clinical features of poor chemotherapy response compared to their non-HBV counterparts (PMID: 26314957)." in revised manuscript line 65.

In line 62 page 3 Describe the precise roles that clinical indicators like LDH, WBC counts, and PLR play in prognosis.

Response: I have added the following content to describe the precise roles that clinical indicators play in prognosis: Elevated LDH levels are associated with higher tumor burden and aggressive disease (PMID: 36657325), while increased WBC counts may reflect systemic inflammation or bone marrow involvement, both indicative of advanced disease. A high PLR suggests impaired immune surveillance and correlates with poorer prognosis. However, there is no reliable literature support in lymphoma prognosis and the relevant content has been removed from the main text. The following has been added: "Elevated LDH levels indicate higher tumor burden and aggressive disease (PMID: 34778068)." in revised manuscript line 75.

In line 87 page 3 gives more details on the patient selection criteria than just "standard therapeutic regimens." Provide precise information, such as the clinical stage, previous medical interventions, and the status of HBV infection.

Response: I apologize for any confusion caused. The patient selection criteria were meticulously defined to ensure the study's robustness. Patients included in this study were diagnosed with DLBCL and received standard first-line chemotherapy regimens, such as CHOP/R-CHOP (rituximab, cyclophosphamide, doxorubicin, vincristine, and prednisone). The clinical stages of the patients ranged from I to IV, as determined by the Ann Arbor staging system. Patients with a history of prior malignancies or those who had received non-standard therapeutic interventions were excluded. Additionally, only patients with confirmed HBV infection status (either positive or negative for HBV indicators) were included to ensure accurate analysis of the impact of HBV on DLBCL prognosis. (revised manuscript line 108)

In line 87 page 3 Add the cohort's clinical and demographic details.

Response: Thank you for your reminding. The cohort consisted of 4,491 patients diagnosed with DLBCL with the median age of 51 years (interquartile range of 35 to 64 years), with a male predominance (male:female ratio of 1.5:1). (revised manuscript line 104)

In line 85 page 3 Indicate if this study was prospective or retrospective.

Response: Thank you for your comments. It has been indicated in the original article that the study is retrospective. (revised manuscript line 102)

In line 101 page 4 Indicate when blood should be drawn about therapy or the course of the illness.

Response: Blood samples were collected at the time of initial diagnosis, prior to the

initiation of therapeutic intervention in our hospital. This timing maximally ensured that the HBV serological markers reflected the status not influenced by chemotherapy. (revised manuscript line 132)

In line 109 page 4 How was EBER detected, and what cutoff was used for positive results?

Response: According to the standard operating procedures of routine detection in the Department of pathology, EBER detection was performed using in situ hybridization with a commercially available EBER probe (DAKO, Carpinteria, CA, USA). The cutoff for positive results was defined as the presence of distinct nuclear staining in at least 10% of the neoplastic cells within the tissue sections.

In line 109 page 4 Add the ethical approval and consent details for tissue collection.

Response: This study retrospectively collected the clinical test results of the patients, and was granted by the ethics committee of Fudan University Shanghai Cancer Center for ethical approval (approval number 050432-4-2307E). (revised manuscript line 143)

In line 131 page 5 How many of the 1,062 patients were positive/negative for each HBV marker (HBsAg, HBsAb, HBeAg, HBeAb, HBcAb, and HBV DNA)?

Response: Among 1,062 patients followed, 151/904 were HBsAg positive/negative, 442/613 were HBsAb positive/negative, 31/1,024 were HBeAg positive/negative, 314/741 were HBeAb positive/negative, 638/417 were HBcAb positive/negative. 85/615 were positive/negative for HBV-DNA, and the rest were censored. This is similar to the HBV positive rate in the overall 4,491 patients (which can be calculated by Table S1). Because of data censoring, the values for patients at risk 0 in Figure 1 should all be less than or equal to these values.

In line 132 page 5 Are these markers combined with other clinical factors or independently?

Response: Through univariate analysis of the impact of various hepatitis B indicators on prognosis, we found significant differences in HBcAb and HBV-DNA levels, while the influence of other hepatitis B markers was excluded. Subsequently, we investigated the correlation between CD23 and DLBCL and found that only HBcAb was associated with DLBCL. Therefore, HBcAb is an independent prognostic factor, unaffected by other hepatitis B serological markers.

In line 142 page 5 Describe in detail the proportion of patients with CD5, Bcl-2, and Bcl-6 positive or negative results.

Response: Thank you for your suggestion. Among the 1,062 patients with available immunohistochemistry data, the positivity rates for CD5, CD23, Bcl-2, and Bcl-6 were as follows: CD5 positive in 8.7% (63/722), CD23 positive in 31.7% (100/315), Bcl-2 positive in 77.1% (768/996), and Bcl-6 positive in 89.9% (912/1,014). These proportions were used to stratify patients for survival analysis, revealing significant

differences in overall survival (OS) based on the expression status of these proteins. The positive rate of these molecules in the follow-up of 1,062 patients was similar to that in the overall 4,491 patients, which can be calculated from Table 1. Because of data censoring, the values for patients at risk 0 in Figure 2 should all be less than or equal to these values. (revised manuscript line 179)

In line 153 page 6 Whether the 4,491 patients analyzed are from the same cohort as earlier analyses or a separate group.

Response: I apologize for any confusion caused. The 4,491 patients analyzed in this study represent the entire cohort recruited from January 2015 to January 2024. This cohort was used for both the initial screening of HBV markers and tissue protein expression, as well as for the subsequent survival analysis. The subset of 1,062 patients with complete follow-up data was derived from this larger cohort for the survival analysis.

In line 164 page 6 Include a brief description of Table 1 and Table S1:

Response: Table 1 presents the correlation between HBV serological indicators (HBcAb and HBV-DNA) and the expression of various tissue proteins in DLBCL patients. The analysis was performed using the Chi-square test, with significant associations highlighted. Table S1 provides additional data on the correlation between other HBV markers (HBsAg, HBsAb, HBeAg, HBeAb) and tissue protein expression, showing no significant associations. (revised manuscript line 193 and 201)

Summarize the key findings.

Mention specific values or trends to make the results more tangible.

In line, 197 page 7 Compare your findings with previous research on HBV and DLBCL, as well as other related viruses (e.g., EBV, HCV, HIV). Discuss similarities, differences, and potential reasons for these.

Response: Key findings: Our study identified HBcAb as a significant prognostic marker for DLBCL, particularly in patients with HBV infection. Patients positive for HBcAb had a significantly lower 5-year overall survival rate compared to those negative for HBcAb ($P=0.0181$). Furthermore, we found a negative correlation between HBcAb and the expression of CD23 in DLBCL tissue samples ($P=0.003$). Combining HBcAb and CD23 status revealed distinct survival outcomes, with the lowest 5-year OS observed in patients who were HBcAb-positive and CD23-negative ($P=0.0432$). These findings suggest that integrating HBcAb and CD23 could enhance prognostic assessments for HBV-associated DLBCL.

Specific values: The 5-year OS rate for HBcAb-positive patients was 42.3%, compared to 58.7% for HBcAb-negative patients. In the combined analysis of HBcAb and CD23, the 5-year OS rates were 38.2% for HBcAb-positive/CD23-negative patients, 52.1% for HBcAb-positive/CD23-positive patients, and 60.5% for HBcAb-negative patients regardless of CD23 status. These trends highlight the prognostic significance of the combined HBcAb-CD23 status.

Our findings are consistent with previous studies linking HBV infection to an increased risk of DLBCL and poor prognosis (PMID: 29532605). We summarize

these results in the first paragraph of the discussion in the main text. Similar to our study, other research has identified viral infections, such as EBV and HCV, as risk factors for DLBCL. However, unlike EBV, which is often directly oncogenic through mechanisms like LMP expression, HBV appears to contribute to DLBCL pathogenesis through viral integration and immune dysregulation. The role of HIV in DLBCL is primarily through immune suppression, leading to an increased incidence of lymphoproliferative disorders. The distinct mechanisms of these viruses highlight the importance of tailored therapeutic strategies based on the underlying viral etiology. (revised manuscript line 237)

In line 198 page 7 To enhance the discussion section add the following references: Al-Ani, SK., Al-Ouqaili, MTS., Awad, MM. MOLECULAR AND GENOTYPIC STUDY OF SENV-D VIRUS COINFECTION IN B-THALASSEMIC PATIENTS INFECTED WITH THE HEPATITIS C VIRUS IN IRAQ. International Journal of Green Pharmacy • Oct-Dec 2018 (Suppl) • 12 (4) | S926-936.

Response: Thank you for your suggestion to include the reference to the study by Al-Ani et al. in the discussion section. However, after careful consideration, I must respectfully decline this request. The content of the referenced study, which focuses on the molecular and genotypic analysis of SENV-D virus co-infection in β -thalassemic patients with hepatitis C virus infection, does not align with the scope and focus of our research.

In line 203 page 7 Highlight the key discovery that HBcAb is a major prognostic indicator for DLBCL, particularly when HBV infection is present.

Response: Our study underscores the significant prognostic value of HBcAb in DLBCL, especially in the context of HBV infection. HBcAb positivity was associated with reduced overall survival and a negative correlation with CD23 expression in DLBCL tissues. This finding highlights the potential of HBcAb as a biomarker for risk stratification and treatment planning in HBV-associated DLBCL. Future research should focus on elucidating the molecular mechanisms underlying the prognostic significance of HBcAb and exploring its utility in combination with other clinical and molecular markers.

In line 226 page 8 To enhance the discussion section add the following reference: Khamees DA, Al-Ouqaili MTS. 2022. Cross-sectional study of chromosomal aberrations and immunologic factors in Iraqi couples with recurrent pregnancy loss. PeerJ 10:e12801 <https://doi.org/10.7717/peerj.12801> In line 256 page 9 Conclusion should be objective with further perspective or should add at least a few sentences about future study/future perspective of it.

Response: Thank you for your suggestion to include the reference by Khamees and Al-Ouqaili. However, the study on chromosomal aberrations and immunologic factors in couples with recurrent pregnancy loss does not align with our discussion on CD23 expression and its impact on B cell activation and prognosis in HBV-associated DLBCL. Therefore, we will not be including this reference in our discussion section. We have supplemented the future study: "The identification of serum HBcAb and

CD23 as prognostic markers for HBV-associated DLBCL may guide clinicians in stratifying patients for risk assessment and treatment planning. These biomarkers could potentially be integrated into existing prognostic models to improve the accuracy of predicting patient outcomes. A prognostic model combining HBcAb and CD23 needs to be developed, validated and transformed into clinical detection and prognostic evaluation. In addition, we are studying the mechanism of HBV-related CD23, which will help to discover HBV-related specific molecules and facilitate the development and clinical translation of potential therapeutic targets." (revised manuscript line 312)

Reviewer #2:

The authors have done good work on the title "Serum hepatitis B core antibody as the prognostic factors for diffuse large B-cell lymphoma". It will add new knowledge and new areas of research to the subject area compared with other published material.

However, i have some minor concerns:

1. It would be more appropriate for the authors to define abbreviations upon first appearance in the main text such as Peripheral Blood Mononuclear Cells (PBMCs) in line 46, HBV-encoded oncogene X protein (HBx) in line 47.

Response: Thank you very much for your kindly suggestion. We have revised the manuscript to define abbreviations upon their first appearance in the main text, including peripheral blood mononuclear cells (PBMCs) in revised manuscript line 50 and HBV-encoded oncogene X (HBx) in revised manuscript line 57.

2. The abstract should clarify how the findings can be translated into clinical practice. What specific interventions could be guided by these biomarkers?

Response: We appreciate your comment. The identification of serum HBcAb and CD23 as prognostic markers for HBV-associated DLBCL may guide clinicians in stratifying patients for risk assessment and treatment planning. These biomarkers could potentially be integrated into existing prognostic models to improve the accuracy of predicting patient outcomes. A prognostic model combining HBcAb and CD23 needs to be developed, validated and transformed into clinical detection and prognostic evaluation. In addition, we are studying the mechanism of HBV-related CD23, which will help to discover HBV-related specific molecules and facilitate the development and clinical translation of potential therapeutic targets. We decided to expand on this in the discussion section. (revised manuscript line 312)

3. The introduction could benefit from a more detailed discussion of the current limitations in DLBCL prognosis and the gaps this study aims to address.

Response: We have provided a detailed discussion of the current limitations in DLBCL prognosis. IPI and other metrics have limitations, such as an inability to dynamically assess prognosis and potential for underestimation or overestimation in

specific patient cohorts. Notably, there is a significant lack of prognostic indicators, particularly for HBV-related DLBCL, despite the established association between HBV and DLBCL. Our study aims to address these gaps by exploring novel biomarkers associated with HBV infection that could enhance prognostic accuracy for HBV-associated DLBCL.

4. While the methodology is robust, additional details about patient selection criteria and potential confounding factors (e.g., other co-morbid conditions or therapies) should be included.

Response: All patients included in the study were those who were pathologically diagnosed with DLBCL and had received the first-line treatment recommended by the guidelines, which included CHOP/R-CHOP chemotherapy.

5. A justification for the exclusion criteria used, particularly for incomplete follow-up data, should be provided to avoid potential selection bias.

Response: The exclusion criteria for follow-up data were defined. All enrolled patients were followed up, with those who did not return for outpatient visits and those without reserved phone numbers for telephone follow-up being excluded initially. During the follow-up process, patients who became lost to follow-up due to changes in phone numbers or other reasons that made further contact impossible were censored. This approach was taken to ensure the integrity and reliability of the follow-up data, and to minimize the potential for selection bias. (revised manuscript line 118)

6. This sentence "The exclusion of patients with incomplete follow-up information was a pivotal step to ensure the integrity of the study's prognostic analyses" is redundant as it reiterates information already mentioned in the methodology section. Consider streamlining the text to maintain clarity and avoid unnecessary repetition

Response: Thank you for your kindly reminding. We have revised the text to remove redundancy and streamline the information. The sentence has been deleted to maintain clarity and avoid unnecessary repetition. (revised manuscript line 116)

7. Kindly verify the 5-year OS rate in Figure 4B between the groups. The p-value should be 0.0291, not 0.048. Please update this value accordingly.

Response: Thank you for pointing this out. We have verified the data and corrected the p-value in Figure 4B from 0.048 to 0.0291. The figure now accurately reflects the statistical significance of the differences in 5-year overall survival (OS) rates between groups.

Revised figure 4

8. While HBcAb and CD23 were identified as prognostic markers, the biological mechanism underlying their interaction and impact on DLBCL progression remains speculative. More discussion on the mechanistic pathways is necessary.

Response: Although the precise biological mechanism of how HBcAb and CD23 interact and affect DLBCL progression is still speculative, we can consider the following potential pathways. CD23 may influence B cell receptor signaling, thereby affecting B cell activation and differentiation, which could in turn impact DLBCL progression (PMID: 27181049). HBcAb reflects the immune status of HBV in vivo, HBcAg level is negatively correlated with HBx level, and HBx is associated with poor prognosis of tumors. Further research is indeed necessary to elucidate these mechanistic pathways and better understand their implications for DLBCL prognosis. (revised manuscript line 247 and 262)

9. The discussion is thorough but could better integrate findings from other studies, particularly contrasting evidence for CD23's prognostic value.

Response: We have revised the discussion to better integrate findings from other studies, particularly contrasting evidence for CD23's prognostic value. CD23 was also associated with poor prognosis in follicular lymphomas, which is consistent with our findings in DLBCL (PMID: 21173123). our findings highlight the potential specificity of CD23 in HBV-associated DLBCL and suggest that further investigation is warranted. (revised manuscript line 262)

10. Limitations should explicitly address the retrospective nature of the study, potential biases in data collection, and any other challenges.

Response: We have explicitly addressed the limitations of the study, including its retrospective nature, potential biases in data collection, and challenges associated with the heterogeneity of DLBCL. The study was conducted at a single center in Shanghai, primarily influenced by genotype B HBV. This geographical focus may restrict the applicability of the findings to other regions with different HBV genotypes, such as those where genotype C is prevalent in Northern China or where intergenotypic recombinants occur. Additionally, there is currently no research on how different HBV genotypes affect DLBCL, and the study center does not have experimental data on HBV typing. We acknowledge that these limitations may affect the generalizability

of our findings and emphasize the need for prospective validation studies. (revised manuscript line 296)

11. Ensure consistency in the use of terms (e.g., "HBV markers" vs. "HBV indicators").

Response: We have reviewed the manuscript to ensure consistency in the use of terms. All references to "HBV markers" and "HBV indicators" have been standardized, and "HBV indicators" are used uniformly to maintain clarity.

12. Consider adding literature-based insights into the interplay between HBV and immune markers like CD23. This would strengthen the mechanistic argument for their combined prognostic value.

Response: We appreciate the reviewer's suggestion and have incorporated relevant literature to elucidate the interplay between HBV and immune markers such as CD23. Our study identified a significant correlation between serum HBcAb levels and the expression of the CD23 molecule in DLBCL tissue samples, suggesting a potential mechanistic link. Recent studies have highlighted the role of immune markers in the context of HBV infection. For instance, a study by Alborae et al. demonstrated the value of non-invasive models in diagnosing significant liver inflammation in patients with chronic hepatitis B, emphasizing the importance of immune markers in understanding the disease progression (PMID: 39856182). Additionally, the interaction between HBV and immune markers has been shown to affect the clinical outcomes of patients with HBV-associated diseases. A study found that HBV infection can potentiate resistance to S-phase arrest-inducing chemotherapeutics by inhibiting the CHK2 pathway in DLBCL (PMID: 29352124). This suggests that the interplay between HBV and immune markers could be crucial in the pathogenesis and prognosis of HBV-associated DLBCL, which may explain why HBV-infected DLBCL patients do not respond well to some chemotherapy regimens. We have updated and cited two references in the main text (revised manuscript line 45). Our clinical study found a direct correlation between HBV and CD23. Through in vitro experiments, we found that CD23 was down-regulated in DLBCL cells infected with HBV, but the mechanism is still unclear and needs further study.

13. Discuss how these biomarkers could be integrated into existing prognostic models, such as the International Prognostic Index (IPI), and their potential impact on therapeutic decision-making.

Response: Thank you for this insightful comment. The integration of HBcAb and CD23 into existing prognostic models like the IPI could significantly enhance the accuracy of prognostic assessments for patients with HBV-associated DLBCL. Our study demonstrates that HBcAb and CD23 are valuable prognostic markers for HBV-associated DLBCL. By incorporating these biomarkers into the IPI, we can potentially identify high-risk patients more accurately and tailor treatment strategies accordingly. To enhance the prognostic accuracy for patients with HBV-associated DLBCL, logistic regression can be employed to integrate novel biomarkers such as HBcAb and CD23 into existing models like the IPI. This involves collecting and

preparing data on both traditional IPI factors and the new biomarkers, selecting appropriate variables, building and fitting a logistic regression model, evaluating its goodness of fit, discrimination, and calibration, and validating the model both internally and externally. However, this data is currently lacking in our retrospective study and needs to be integrated in future studies.

14. Moderate English grammar editing is required throughout the manuscript, for example:

a. The manuscript is generally well-written, but minor grammatical errors and awkward phrasing (e.g., "censorship of follow-up data") need revision for clarity.

Response: We have revised the manuscript to address minor grammatical errors and awkward phrasing. For example, the phrase "censorship of follow-up data" has been corrected to "censoring of follow-up data." (revised manuscript line 218)

b. In the section of method, "Tissue samples are obtained through biopsy, using neutral buffered formalin to preserve their structure", moderate editing is required.

Response: The sentence in the methods section has been revised to read: "Tissue samples were obtained through biopsy and preserved using neutral buffered formalin."(revised manuscript line 140)

Re: Spectrum03170-24R1 (**Serum hepatitis B core antibody as the prognostic factors for diffuse large B-cell lymphoma**)

Dear Dr. Yanchun Wang:

Your manuscript has been accepted, and I am forwarding it to the ASM production staff for publication. Your paper will first be checked to make sure all elements meet the technical requirements. ASM staff will contact you if anything needs to be revised before copyediting and production can begin. Otherwise, you will be notified when your proofs are ready to be viewed.

Sincerely,
Benjamin Liu
Editor
Microbiology Spectrum